# Orienting attention to auditory and visual working memory in older adults with cochlear implants

Amisha Ojha [1,2,3] *, Andrew Dimitrijevic[2,3,4,5], Claude Alain[1,3,4,6]

**1** Rotman Research Institute, Baycrest, Toronto, Ontario, Canada, **2** Sunnybrook Health Science Centre, Toronto, Ontario, Canada, **3** Institute of Medical Sciences, University of Toronto, Toronto, Ontario, Canada, **4** Department of Psychology, University of Toronto, Toronto, Ontario, Canada, **5** Department of Otolaryngology–Head and Neck Surgery, University of Toronto, Toronto, Ontario, Canada, **6** Music and Health Science Research Collaboratory, University of Toronto, Toronto, Ontario, Canada

* amisha.ojha@mail.utoronto.ca

## Abstract

Cochlear implantation is a well-established method for restoring hearing sensation in individuals with severe to profound hearing loss. It significantly improves verbal communication for many users, despite substantial variability in patients' reports and performance on speech perception tests and quality-of-life outcome measures. Such variability in outcome measures remains several years after implantation and could reflect difficulties in attentional regulation. The current study assessed the ability to use a cue to guide attention internally toward visual or auditory working memory (i.e., reflective attention) in cochlear implant (CI) users. Participants completed a cognitive task called the delayed match-to-sample task in which a visual gradient was presented on a computer screen and a piano tone was presented through speakers simultaneously. A visual cue (i.e., letter A or V) instructed participants to focus attention on the item held in auditory or visual working memory. After a delay following the cue presentation, participants were presented with a probe item and indicated by pressing a button whether it matched the cued item in working memory. CI users and age-matched normal hearing adults showed comparable benefit from having an informative cue relative to an uninformative cue (i.e., letter X). Although CI users have had a history of severe deafness and experience coarse sound information, they were able to retrospectively orient their attention to an item in auditory or visual working memory. These findings suggest that CI users with at least one year of CI experience can successfully regulate attention to a level that is comparable to that of normal hearing individuals.

## Introduction

Severe-to-profound sensorineural hearing loss can be partially restored using cochlear implants (CI). In adults who acquired hearing loss due to delayed onset or progressive genetic hearing loss, accidents, noise exposure, or viral infections, cochlear implantation is becoming the standard of care. It significantly improves speech perception in older adults [1], though

**Data Availability Statement:** Regrettably, some but not all data could be included in this submission as it contains Personal Health Information, as mandated by the Research Ethics Board at Baycrest Health Sciences. Researchers

who meet the criteria for access to confidential data may do so by contacting the the Research Ethics Board at Baycrest Health Sciences through Noah Koblinsky at nkoblinsky@research.baycrest.org or 416-785-2500 x 2440. Minimum relevant data is available in the Supporting Information files.

**Funding:** This research was supported by grants from the Natural Sciences and Engineering Research Council of Canada (NSERC) Discovery and the William Demant Foundation to Claude Alain [RGPIN- 2021-02721] and NSERC-Create Training Program in Complex Dynamics grant to Amisha Ojha. The funders had no role in study design, data collection and analysis, decision to publish, or preparation of the manuscript. There was no additional external funding received for this study.

**Competing interests:** The authors have declared that no competing interests exist.

there is substantial variability in speech perception and quality-of-life outcome measures. For instance, some CI users report improvements in language development, communication with others, voice recognition, awareness of sounds in the environment and ability to control their own voice [2]. However, other users report negative physical, physiological and emotional experiences that eventually drive their decision to discontinue the use of their implant. A consensus among users is difficulty understanding speech in noisy environments [2, 3]. Such variability in outcome measures remains several years after implantation and could be related to complex interactions between hearing restoration and cognition.

The prevalent view is that hearing restoration through cochlear implantation improves cognition in older adults [4–6]. However, the evidence supporting this assumption is mixed. Some studies report no difference in performance on cognitive measures taken before and after implantation [7, 8]. Castiglione et al. [9] reported a slight improvement on the Montréal cognitive assessment (MoCA) after cochlear implantation. There was no control group in that study, and the same MoCA test was used twice, making it difficult to rule out the practice effect. Other studies showed some improvement in performance at various cognitive tests after cochlear implantation [4–6, 10–13]. Still, these benefits were often limited to those individuals who experienced mild cognitive deficits before cochlear implantation. Lastly, the cognitive domain positively impacted by cochlear implantation varies substantially among the studies, with some studies reporting benefits in short-term, long-term, and working memory tasks [5, 9, 14], while others report benefits in attention [4], processing speed and overall cognition [4, 15].

The sensory-deprivation theory [16, 17] posits that cognitive abilities begin to decline due to the degradation of auditory input associated with hearing impairment. One open question is whether hearing rehabilitation by the use of a CI could mitigate the effects of auditory deprivation on cognition. Studies have examined the effectiveness of CIs in minimizing this effect and provide evidence for CI and hearing aid use resulting in greater cognition than those with hearing loss with no hearing device [18, 19]. However, other studies have not found a positive effect of hearing aids [20] or CIs on cognition [10, 13]. A possible explanation for these inconsistencies in the literature could be related to the degraded sound input of the CI and the continued cross-modal neuroplasticity observed in CI users. The effects of severe to profound deafness on the auditory cortex may remain even after hearing interventions with a CI and perhaps this continued impoverished auditory system is not capable of reversing the negative effects of hearing loss on cognition [21, 22]. The sensory-deprivation theory predicts that the relationship between hearing loss and cognition can be alleviated once hearing is restored.

Age-related hearing loss (ARHL), also known as presbycusis, is a progressive sensorineural hearing loss. Presbycusis has direct effects on communication, quality-of-life and several studies suggest that hearing loss is associated with decline in several cognitive domains including memory, attention, and speed of information processing [23]. Garami et al. [24] adapted a visual and auditory delayed match-to-sample task to examine whether age and presbycusis are associated with deficits in reflective attention. In this paradigm, participants were presented with a stimulus array of four different digits, two of which were presented visually (one in each hemifield) and two aurally (one in each ear). An informative or uninformative (neutral) retro-cue was presented during the retention interval to orient attention towards either the auditory or visual working memory. Older adults with normal hearing and those with presbycusis demonstrated benefit from informative auditory and visual retro-cues. However, benefit of the informative auditory retro-cue varied as a function of hearing status. Specifically, greater hearing loss was associated with reduced response time benefit when attention was cued to the auditory items in working memory. This implies that there may be a link between hearing loss and reflective attention. Older adults with hearing loss may be allocating more attentional

resources to external auditory stimuli, decreasing the cognitive resources available to orient attention towards internal auditory information in working memory. Hearing loss may be reducing the effectiveness of retro-cues in facilitating this internal focus of attention, shown by the slowing of response [24].

One open research question is how hearing restoration with a CI impacts the regulation of attention and memory processes for both visual and auditory material. For instance, severe to profound hearing loss may negatively impact the development or utilization of working memory processes, such as reflective attention. Specifically, the natural inclination to mentally replay auditory experiences, often referred to as "listening back in time," may be diminished in adults with severe to profound hearing loss. This could be attributed to the absence or degradation of sound object representations. Evidence suggests that mild to moderate hearing loss is associated with deficits in retrospectively orienting attention and selecting an item in working memory [24]. This finding implies that hearing impairment may predict deficits in shifting attention to internal auditory representations. Consequently, one would anticipate that CI users would experience difficulties in retrospectively allocating attention to sound object representations in working memory.

Investigating auditory reflective attention in CI users could shed light on possible cognitive outcomes of hearing restoration, specifically attentional processes. Considering the sensory deprivation theory [16, 17], hearing restoration should reduce the draw on cognitive resources, which could be redistributed to other perceptual and cognitive domains. The current study examines whether older CI users could use a visual cue to orient their attention to a visual or an auditory item held in working memory. We measured response times and accuracy scores during a variant of the delayed match-to-sample paradigm that included a retro-cue during the retention interval, directing attention to visual or auditory working memory. We predicted that compared to age-matched controls, CI users will experience more difficulties in focusing and selecting items from auditory than visual working memory (i.e., reflective attention).

## Methods

### Participants

Thirty-two CI users and thirty-eight normal hearing controls were recruited between February 12, 2022, and May 4, 2023, from the patient population in the Department of Otolaryngology at Sunnybrook Health Science Centre and the participant database at the Rotman Research Institute, Baycrest Hospital. Five controls were excluded due to unrecorded responses, programming errors and a history of mild cognitive impairment. Four CI users were also excluded, of which were due to unrecorded responses, and study withdrawal. Regarding the unrecorded responses, it was observed that participants whose response exceeded the timed window for more than ten trials also performed at chance level. Therefore, participants who had ten or more missed trials were excluded from further analysis.

The final sample comprised of twenty-eight CI users aged between 60 and 86 years old (M (mean) = 72.1, SD (standard deviation) = 6.7; 21 males) (S1 Table) and thirty-three age-matched normal hearing controls between 60 and 86 years old (M = 71.6, SD = 7.7; 12 males) (S2 Table). The CI group consisted of eight bilateral CI users and twenty unilateral CI users, of which fourteen used a hearing aid on the contralateral ear. All CI users had at least one year of implantation. A priori power analysis was conducted using G*Power version 3.1.9.6 to estimate the required sample size based on a large effect size of .8 from Alain et al. [25], which compared auditory reflective attention between young and old adults. The minimum total sample size needed to achieve 80% power with a significance criterion of α = .05 for this large effect size was N = 52. We used a conservative approach and anticipated a smaller effect size in

**Table 1. Age and assessment information.**

| | CI (N = 28) | | | NH (N = 33) | | |
|---|---|---|---|---|---|---|
| | M | SD | Range | M | SD | Range |
| Age | 72.14 | 6.7 | 60–86 | 71.56 | 7.73 | 60–86 |
| MoCA | 24.93 | 2.68 | 17–28 | 27.39 | 1.54 | 24–30 |
| QSIN | 15.3 | 4.24 | 9–25.5 | 2.3 | 1.83 | -1-6.25 |

MoCA, Montreal Cognitive Assessment
QSIN, Quick Speech in Noise

the present study and, therefore recruited 61 participants to test the study hypothesis. Informed written consent was obtained from participants in accordance with the guidelines established by the Research Ethics Board at Baycrest Health Sciences and Sunnybrook Health Sciences Centre. All participants received monetary compensation and parking passes at the respective hospital campus.

All participants completed hearing assessments and a behavioral task in a sound-attenuated booth. The hearing assessments included pure tone audiograms (PTA) for octave frequencies from 250–8000 Hz, and four-word lists in the Quick Speech-in-Noise (QSIN) test [26]. Only the NH group completed the PTA, whereas all participants were assessed with the QSIN. All but seven controls showed normal hearing thresholds in the lower frequencies (250–2000 Hz) with an upper cut-off of 20 decibel (dB) hearing level (HL) but had threshold shift to 50 dB HL at frequencies 4000–8000 Hz. These levels are consistent with normal age-related hearing loss at higher frequencies (4000–8000 Hz) as suggested by the American Speech-Language-Hearing Association [27]. All controls were administered the Montreal Cognitive Assessment (MoCA) [28], and CI users completed the Hearing-Impaired MoCA (HI-MoCA) [29], which were used to measure how general cognitive functioning is related to visual and auditory reflective attention. Age and assessment scores of all participants are shown in Table 1. The visual acuity of all participants was quantified as normal or corrected to normal using the Snellen Eye Chart [30].

## Stimuli

The stimuli used in the study comprised of black and white 9 cm Gaussian gradients within a gray rectangle with a height of 16.5 cm and width of 22 cm, auditory piano tones presented at 75 dB sound pressure level (SPL), and white visual letters A, V, or X (Arial font) with the height of 2.5 cm. Gaussian gradients and the letters were presented at the center of a computer screen, while piano tones were presented binaurally through two speakers placed at a 45-degree angle from the participant. All piano tones were generated by Audacity Software, and Gaussian gradients ranging from an angle orientation of 5 to 360 degrees were created from the Visual Stimulus Generation Toolkit on Neurobehavioral Presentation Software (Version 16.3, www.neuro-bs.com). All stimuli were presented on a computer screen with a black background.

## Experimental procedure

**Familiarization phase.** Prior to beginning the experiment, all participants completed two simplified unimodal tasks. These tasks were used to familiarize the participants with the main task and to ensure a level of accuracy above chance and between 85–90% by adjusting the level of discriminability between the different stimuli.

This familiarization phase aimed to minimize the expected difference in pitch acuity between CI users and age-matched controls, so CI users and controls would be presented with stimuli that were closely matched in terms of their discriminability.

In the auditory delayed match-to-sample task, participants were first shown a white fixation cross for 1000 ms, followed by a 500 ms piano tone, a 1000 ms blank screen as a delay, followed by a second presentation of a 500 ms piano tone immediately following the first. Participants indicated whether the tones were identical or different by pressing the right or left arrow key, respectively, on a computer keyboard. The visual delayed match-to-sample task followed the same procedure with Gaussian gradients rather than piano tones. Each unimodal task consisted of 14 trials.

The physical difference between the gradients (i.e., visual angle rotation) and piano tone (i.e., frequency) varied between versions of the tasks. The difference in visual angle between gradients ranged from a random rotation of 25 to 60 degrees for the visual delayed match-to-sample task. For the auditory delayed match-to-sample, the difference in frequency varied from one-quarter tone to seven semitones.

All participants began the familiarization phase with a difference of a 30-degree visual angle rotation and two semitones for the visual and auditory delayed match-to-sample, respectively. They completed a block of 14 trials. If performance was lower than 85%, the physical difference was increased. If performance was at the ceiling (i.e., 100% accuracy), then the difference was reduced. This process was repeated until performance was approximately 85% accuracy in both auditory and visual working memory tasks. For each participant, the auditory and visual stimuli that yielded approximately 85% accuracy in the unimodal task was then incorporated into the audio-visual delayed match-to-sample task.

## Audiovisual delayed match-to-sample

In the audio-visual delayed match-to-sample paradigm, participants were first shown a white fixation cross on a black background for 1000 ms, followed by a brief presentation of a piano tone and visual gradient simultaneously for 500 ms. Once the stimuli disappeared, a black screen was shown for 1000 ms, followed by another 500 ms presentation of a letter retro-cue that was either neutral (X) or informative (A or V to indicate whether to orient to auditory or visual modality, respectively). In trials with the informative A retro-cue, participants were tasked to remember the piano tone, whereas in trials with the informative V retro-cue, participants were instructed to remember the visual gradient. After another brief 2000 ms black screen, a unimodal test item was presented for 500 ms, and participants indicated whether the item matched with the auditory or visual stimuli held in their working memory (Fig 1). For instance, a trial with an informative A retro-cue would be followed by a piano tone as the test object, and a trial with an informative V retro-cue would be followed by a visual gradient as the test item. In the neutral X retro-cue condition, participants were required to retain both auditory and visual items in memory, as no indication was provided regarding which modality would be tested. The test item could originate from either modality, necessitating participants to match it with the corresponding item in their working memory. This condition accounted for 50% of the trials and required participants to be prepared for either test item.

The outcome measures were response time (RT) and accuracy. To retain as much data as possible for analysis while limiting bias of differing sample size, we used the non-recursive moving criterion method to remove outliers in RT [31]. This method removes RT data that falls outside a cut-off value determined by the standard deviation of a given sample size. The standard deviation used for the cut-off is thus altered based on the sample size.

The task was composed of 144 trials, divided into three blocks of 48 trials. Each block took approximately seven minutes to complete. Within a block of 48 trials, 50% were neutral (X) retro-cue. The remaining informative retro-cue trials were equally divided into auditory (A) and visual (V) retro-cues. The trial type (X, A, or V) occurred in random order.

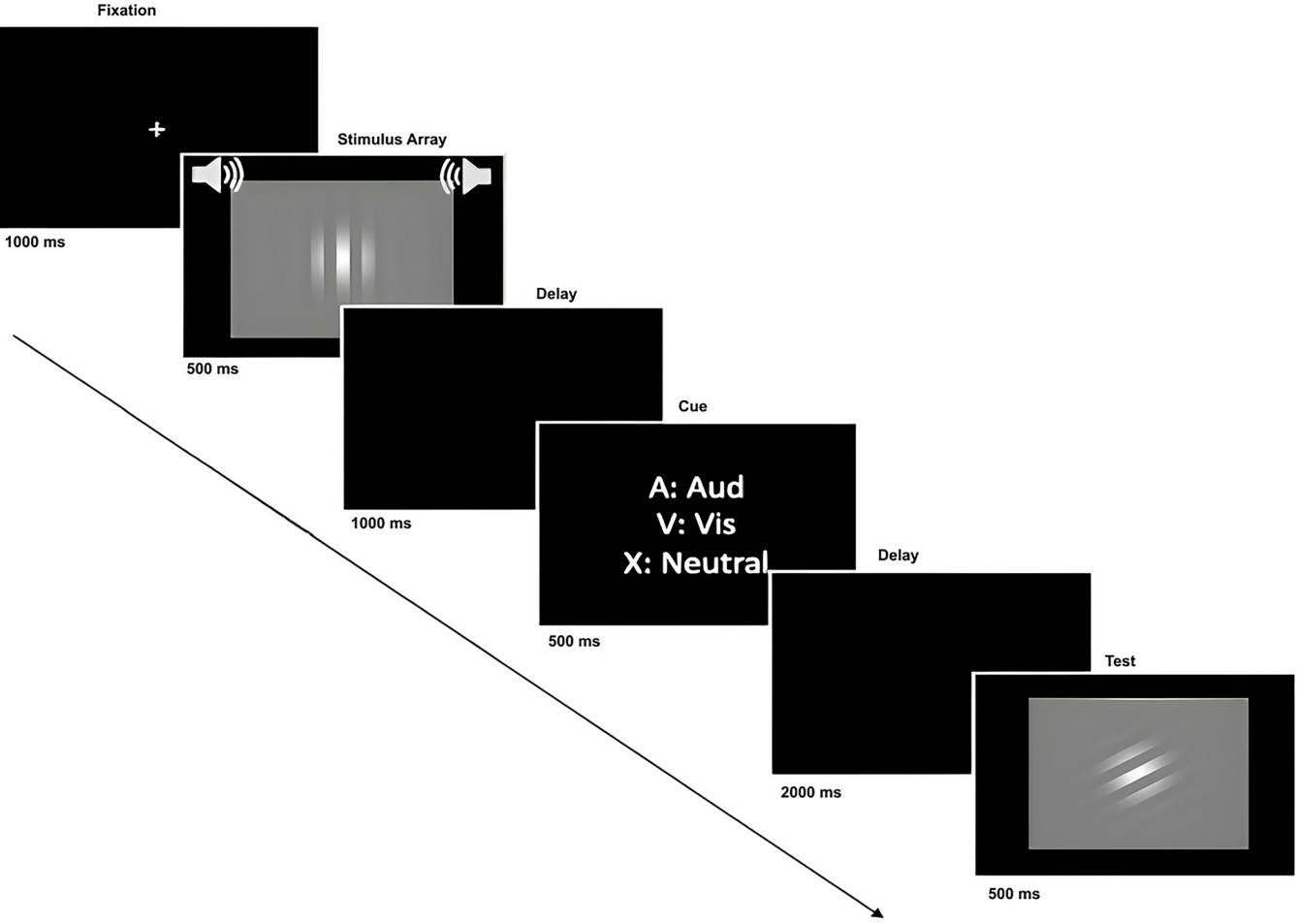

**Fig 1. Schematic of a trial in the audiovisual delayed match-to-sample task.** On each trial, participants were first presented an image of a Gaussian gradient simultaneously with a sound of piano tone presented through two loudspeakers (placed at about an angle of 45 degrees relative to participants' head) at a 65 dB SPL. After a delay, one of three retro-cues was presented visually to either orient attention to auditory, visual or neither working memory modalities. After another delay, a probe item was presented, and participants were tasked to indicate whether or not the probe was identical to the stimulus presented earlier.

Retro-cue benefit was calculated using RT data by comparing the benefit or cost of having an informative retro-cue relative to the neutral retro-cue. This was done for each participant using the following formula:

$$RT\ Benefit = \frac{RT_{neutral} - RT_{informative}\ (auditory\ or\ visual)}{RT_{neutral}} \times 100$$

Performance in the audiovisual task was represented by the calculation of the d-prime (d') value, a measure of discriminability derived by the signal detection theory [32]. The responses were labeled as either a hit, miss, false alarm or a correct rejection, in identifying whether the probe was identical or different than the item in the memory array. D-prime value was the difference between the z-transformed hit and false alarm rates.

## Statistical analysis

RT and accuracy measures were subjected to mixed model analysis of variance (ANOVA), with condition (informative vs. neutral) and modality (visual vs. auditory) as within-subject

factors and group (CI vs. NH) as a between-subject factor. For all follow-up pairwise comparisons, the p-value was corrected using the Bonferroni adjustment for multiple comparisons. All ANOVAs and correlations were two-tailed and the alpha criterion for Type I error was set at 0.05. All statistical analyses were run using IBM SPSS Statistics Version 28 and R Version 4.2.2.

## Results

### Cognitive and hearing assessments

Fig 2 shows the group mean performance on the MoCA and QSIN tests. There was a significant difference between groups on MoCA scores, t(59) = -4.43, p < .001, Cohen's d = -1.138, with controls (M = 27.39, SD = 1.53) scoring higher than CI users (M = 24.96, SD = 2.67). Controls (M = 2.31, SD = 1.86) also performed significantly better (lower threshold) than CI users (M = 14.92, SD = 3.81) in the QSIN task, t(58) = 16.75, p < .001, Cohen's d = 4.346.

### Familiarization phase

On average, CI users needed a larger difference in pitch to reach the performance criteria compared to NH participants, t(59) = 9.054, p < .001 (Fig 3). The difference in visual angle needed to reach the performance criteria was comparable in NH and CI users, t(59) = 1.827, p = .073.

We also examined whether the two groups differed in accuracy, which was defined as the proportion of correct responses from the last sequence used in the familiarization phase and included both hits and correct rejections. The analyses revealed a significant interaction between group and modality, F(1, 57) = 6.899, p = .011, $\eta^2$ = .108 (Fig 4). Despite our attempt to adjust for perceptual difference in auditory acuity and equate performance between groups, CI users were less accurate (M = .862) than controls (M = .94), p < .001. For the visual modality, there was no significant difference in accuracy between CI users and age-matched controls.

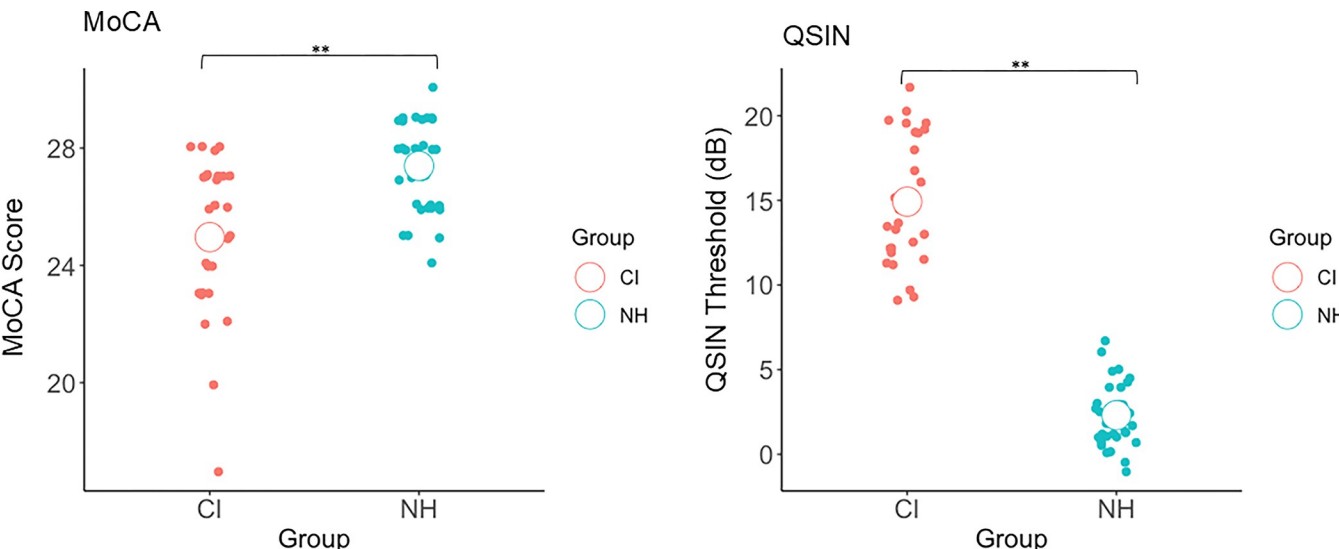

**Fig 2. MoCA scores and QSIN thresholds of cochlear implant (CI) users and normal hearing controls (NH).** MoCA scores between CI users and controls were significantly different. QSIN threshold of CI users and controls were also significantly different. Single dots represent the data points of each score. White circle represents the mean score of each group. **p < .001.

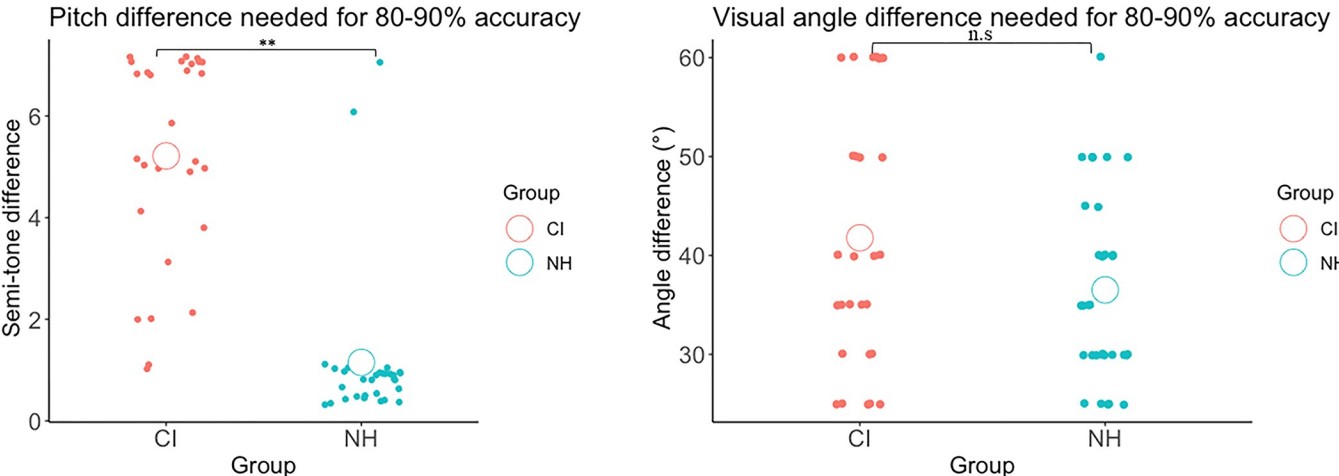

**Fig 3. Level of semi-tone and visual angle difference required for high accuracy in both CI users and NH.** Group mean difference in pitch and visual rotation to attain the accuracy criteria. White circle represents the mean score of each group. CI users needed a larger semitone difference to acquire a high level of accuracy. **p < .001, n.s. = non-significant.

## Performance during the audiovisual delayed-match-to sample task

The percent correct data did not have a normal distribution. We used D-prime values to assess the effect of group and retro-cue on accuracy. RT data were also not normally distributed and therefore were log-transformed before statistical analyses.

The mixed model ANOVA with a Greenhouse-Geisser correction was conducted with group as the between subject factor, stimulus modality and retro-cue type as the within subject factors and D-prime as the dependent variable. This analysis revealed a main effect of retro-cue type on D-prime, $F(1, 58) = 17.79$, $p < .001$, $\eta^2_p = .235$. Overall, participants were more accurate for informative (M = 2.695) than neutral (M = 2.337) cues. Participants also showed higher accuracy for auditory (M = 2.693) than visual (M = 2.339) stimuli, $F(1, 58) = 7.509$, $p = .008$, $\eta^2_p = .115$. The main effect of group on D-prime was also significant, $F(1, 58) = 4.649$, $p = .035$, $\eta^2_p = .074$, where controls (M = 2.698) were overall more accurate than CI users (M = 2.334).

Notably, the interaction between stimulus modality and group was significant, $F(1, 58) = 12.273$, $p < .001$, $\eta^2_p = .175$ (Fig 5A). Pairwise comparisons showed that controls (M = 3.121) were more accurate than CI users (M = 2.285) for auditory stimuli, $p < .001$ (Fig 5A). There was no significant difference between the two groups in accuracy for visual stimuli, $p = .601$. In the controls, accuracy was also higher for auditory than visual stimuli ($p < .001$), whereas no such difference was observed in the CI users ($p = .603$). Finally, the three-way interaction between group, stimulus modality and retro-cue type on accuracy was not significant, $F(1, 58) = 1.728$, $p = .194$, $\eta^2_p = .029$.

Given that CI users required greater pitch difference to reach performance criteria for the audiovisual delayed match-to-sample task, we conducted an analysis of covariance (ANCOVA) to determine if the difference in sensitivity between groups in the auditory modality was influenced by participants' perceptual discrimination ability. The difference between the two groups was not significant, $F(1, 58) = 1.97$, $p = .166$, $\eta^2_p = .033$ (Fig 5B) once semitone difference between groups was taken into account.

For the RT data, a similar mixed model ANOVA showed a main effect of retro-cue type, $F(1, 59) = 157.299$, $p < .001$, $\eta^2_p = .727$ (Fig 5C), with faster RT for informative than neutral cues. There was also a main effect of group, $F(1, 59) = 6.485$, $p = .014$, $\eta^2_p = .099$ (Fig 5D),

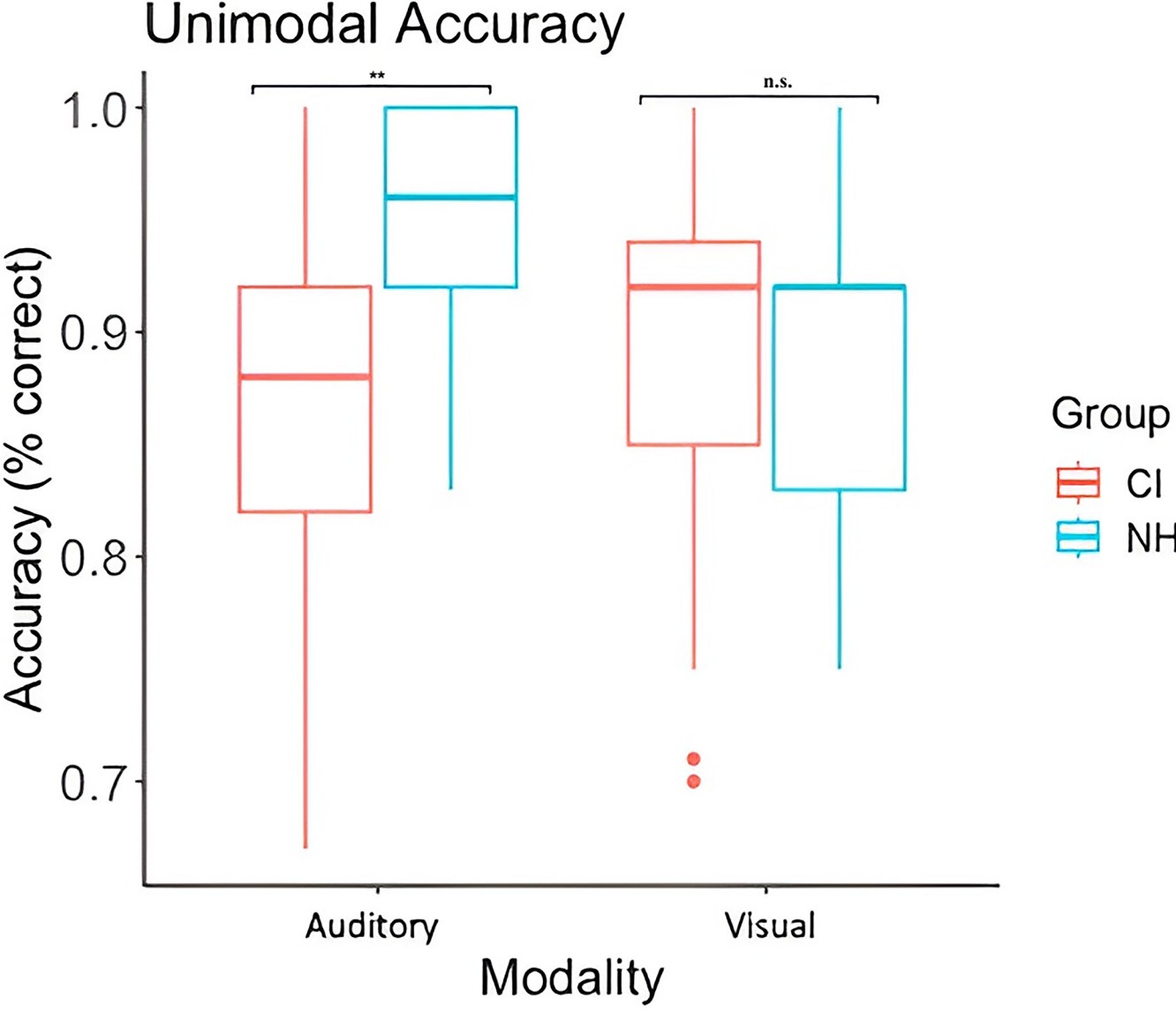

**Fig 4. Performance of each participant group in the unimodal auditory and visual delayed match- to-sample tasks.** All participants performed above chance level. Box represents the interquartile interval where 50% of the data lies. Lines in boxplot represent median accuracy score in each modality. $^{**}$p < .001, n.s. = non-significant.

where controls responded faster than CI users. The main effect of modality was not significant, F(1, 59) = .05, p = .823, $\eta^2_p$ = .001 (Fig 5C). The main effect of group was no longer significant, F(1, 58) = .038, p = .856, $\eta^2_p$ = .001, once the semitone difference between groups was controlled (Fig 5D).

There was a significant interaction between group and modality, F(1, 59) = 8.738, p = .004, $\eta^2_p$ = .129. Pairwise comparisons showed that controls were quicker to respond to auditory conditions than CI users (p = .008), whereas no difference between the groups was observed for visual stimuli (p = .233). Normal hearing controls were also faster for auditory than visual stimuli, p = .022 (Fig 5C). CI users showed no significant difference in RT between the two modalities, p = .068. The group by retro-cue type interaction was not significant, F(1, 59) = .576, p = .451, $\eta^2_p$ = .01, nor was the interaction between retro-cue types and stimulus

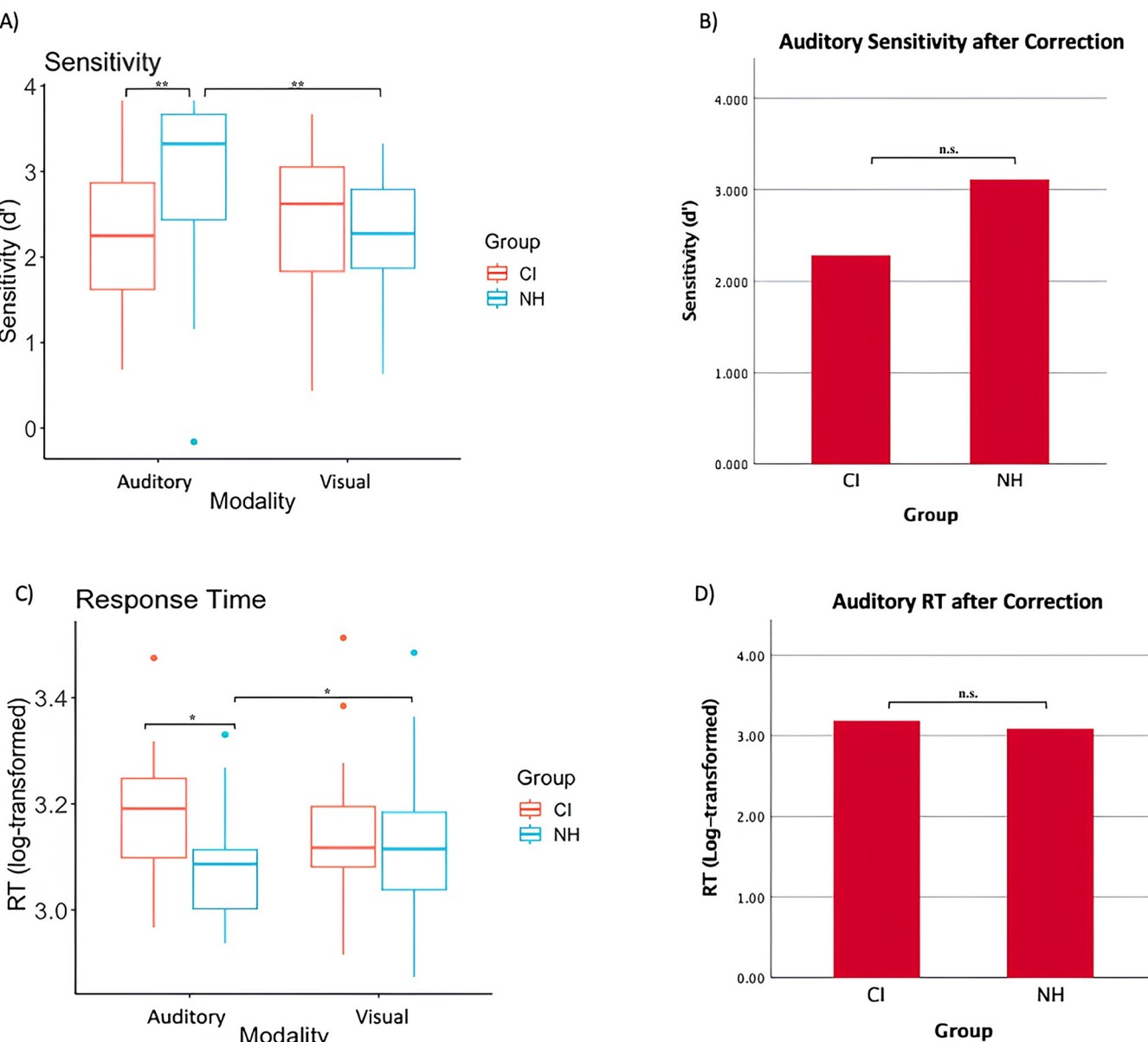

**Fig 5. Task sensitivity and RT between CI users and NH.** A) Group mean D-prime from cochlear implant (CI) users and normal hearing (NH) controls for both auditory and visual modality conditions regardless of cue type. There was a difference between CI and NH in the auditory domain. B) Mean D-prime in the auditory domain was not different between CI and NH when semitone differences were controlled. *p < .05, **p < .001, n.s. = non- significant. C) Group mean response time (RT) (log-transformed) for both auditory and visual modality conditions (informative and neutral). Lines in the box plot represent mean RT for each modality. There was a difference between CI and NH in the auditory domain. D) RT-log in the auditory domain was comparable between CI and NH when semitone differences were controlled. *p < .05, n.s. = non- significant.

modality, $F(1, 59) = 1.151$, $p = .288$, $\eta^2_p = .019$. The three-way interaction between group, retro-cue type, and stimulus modality was not significant, $F(1, 59) = .251$, $p = .6181$, $\eta^2_p = .004$.

## RT benefit of retro-cue

We also examined the effects of retro-cueing on RT benefit using a mixed model ANOVA with group (CI and controls) as the between-subjects factor and modality (auditory and visual) as the within-subjects factor. RT benefit was obtained by comparing the benefit or cost of

having an informative relative to the neutral retro-cue, for both auditory and visual modalities. It minimized group and modality differences in absolute RTs. The mixed model ANOVA with a Greenhouse-Geisser correction revealed no main effect of modality on RT benefit, $F(1, 59)$ = 1.253, $p$ = .268, $\eta^2_p$ = .021. There was no main effect of group on RT benefit, $F(1, 59)$ = .531, $p$ = .469, $\eta^2_p$ = .009, nor was there a significant interaction between group and modality, $F(1, 59)$ = .246, $p$ = .622, $\eta^2_p$ = .004. Altogether, the retro-cue benefit on RTs was comparable between CI users and age-matched controls, and this benefit did not differ between the two modalities.

The effect of biological sex on RT benefit was also examined because there were differences in the number of males and females between the two groups. There was no effect of sex on the overall RT benefit of both visual ($t(59)$ = -1.614, $p$ = .112, Cohen's d = -.415) and auditory ($t(59)$ = -.439, $p$ = .662, Cohen's d = -.113) modalities. Moreover, there was no significant difference in retro-cue benefit on RT between the two types of CI (bilateral or unilateral) for either the visual ($t(26)$ = -1.03, $p$ = .313, Cohen's d = -.413) or auditory ($t(26)$ = -.397, $p$ = .695, Cohen's d = -.166). There was also no significant difference between hearing aid use and no hearing aid for both retro-cue benefit in RT for visual ($t(26)$ = 1.00, $p$ = .326, Cohen's d = .367) and auditory ($t(26)$ = .911, $p$ = .371, Cohen's d = .344) modalities.

## Correlations

Bivariate Spearman's correlations were calculated between CI duration and retro-cue benefit of RT for both auditory and visual domains. All p-values were adjusted using False Discovery Rate (FDR) corrections for multiple comparisons. The correlation between CI duration and retro-cue benefit in RT for either modality was not significant. As an exploratory analysis, further bivariate Spearman's and Pearson's correlations were calculated between age, MoCA scores, QSIN threshold, PTA threshold and retro-cue benefit of RT for both auditory and visual domains. Spearman's correlations were performed for data that were not normally distributed, whereas Pearson's correlations were conducted for normally distributed data. All p-values were adjusted using FDR corrections for multiple comparisons. In CI users, there were no significant correlations between the outcome measures. As for the control group, there was a significant positive correlation between age and PTA threshold (rho = .659, $p$ = .004), a positive correlation between age and QuickSIN scores, ($r(24)$ = .407, $p$ = .041), and a positive correlation between age and retro-cue benefit of RT in the visual domain, ($r(24)$ = .634, $p < .001$).

## Linear regressions

Linear regressions of the significant Pearson's correlations were conducted to further determine if age was a predictor of speech perception and visual retro-cue benefit (Table 2). The analysis indicated that age was a significant predictor of speech-in-noise performance for the NH group ($F(1, 24)$ = .226, $p$ = .008, $R^2$ = .226). Age was also a significant predictor of RT retro-cue benefit in the visual domain for NH ($F(1, 24)$ = 16.12, $p < .001$, $R^2$ = .377). No significant predictors were observed for the CI group.

**Table 2. Summary of linear regressions.**

| Variable | Estimate | Std. Error | t-value | p-value |
|---|---|---|---|---|
| **Predicting QSIN with age** | | | | |
| NH age (year) | 0.12 | 0.042 | 2.881 | .008** |
| **Predicting RT benefit for Visual Retro-Cue with age** | | | | |
| NH age (year) | 1.121 | 0.279 | 4.015 | .0005*** |

**p < .01
***p < .001

## Discussion

The role of age and hearing loss on attention to memory has been previously studied, suggesting that hearing loss, rather than age, may be a predictor of poor auditory reflective attention [24]. The aim of this study was to further explore the association between hearing status and reflective attention by examining reflective attention in CI users. Given the findings of Garami et al. [24], it was anticipated that CI users, who experience poor spectral and temporal sound information, would have greater difficulties orienting attention towards an auditory item in working memory compared to NH listeners.

Surprisingly, the results of the current study showed that CI users, like normal hearing age-matched controls, are able to orient attention towards an item held in auditory and visual working memory. There was no relationship between the number of years of implantation and the ability to orient attention towards either auditory or visual working memory. Although NH listeners responded faster and more accurately when recalling auditory items, the results suggest that CI users take advantage of the informative retro-cues at the same level as those who have normal hearing. Hearing restoration of severe to profound hearing loss by at least one year of CI use yields an ability to attend to items in memory at the normal hearing level.

We found that age was an important factor in predicting speech-in-noise perception performance and the ability to benefit from an informative visually orienting retro-cue. This suggests that ageing results in poor speech perception but greater benefit of a retro-cue facilitating retrospective attention towards visual items in working memory. Additionally, ageing was related to poor hearing threshold in controls, highlighting the relationship between ageing and hearing loss.

Altogether, the results of this study suggest that CI users, after one year of implantation, can retrospectively orient attention toward auditory working memory to the same extent as NH age-matched adults. This is surprising given that CI users' overall performance was lower than controls. The latter is consistent with a more coarse acoustic input despite our attempt to match the perceptual discrimination of auditory stimuli in CI users and controls. Our findings suggest that auditory reflective attention may not be affected by hearing loss, as predicted by previous findings of Garami et al. [24]. However, this discrepancy between the two studies may be related to the difference in the number of items held in working memory (see below).

### Reflective attention to auditory and visual working memory

Considering prior research showing a relationship between hearing acuity and reflective attention towards auditory items in working memory [24], it was predicted that the coarse auditory analysis experienced by CI users would induce greater difficulties in auditory reflective attention. Contrary to this prediction, the current study demonstrated similar reflective attention abilities in both visual and auditory modalities for CI users and NH controls. The overall advantage of informative retro-cues compared to neutral cues is in line with previous research suggesting the facilitative action of retro-cues in diminishing the costs or decay of retrieving items held in working memory [33–39]. The current study findings also add to the existing evidence suggesting that CIs may be able to restore auditory perception and thereby improve some cognitive functions to the level of normal hearing [40]. Considering the degraded sound resolution of CIs, older adults with CI who already experience age-related cognitive decline would further require more cognitive resources for top-down information processing. However, in the case of the present study, older CI listeners were able to rely on contextual cues to recover internal information in the same way as NH older adults. An explanation for this observation could be due to the design of the delayed match-to-sample task. Unlike recent studies examining reflective attention with set sizes of three to four items [24, 41, 42], the

current task only consisted of two items, eliciting a lower level of task difficulty. Furthermore, the retro-cues in the present study reflected attention toward a single item in one of two sensory modalities. In Garami et al. [24], the retro-cues oriented attention towards auditory or visual working memory, which comprised two items. Given that in the present study, there was only a single item in visual or auditory working memory, our paradigm did not require searching through and selecting an item within modality-specific working memory systems. This difference could explain why there was no difference in reflective attention between the two groups. Our findings show that CI users can orient attention towards a relevant modality, but further research is needed to determine whether CI users would also show comparable performance when the task requires searching and selecting an item within modality-specific working memory systems.

There was, however, a general slowing of responses and poor performance pattern in orienting to an auditory item from working memory in CI users compared to normal hearing when differences in discriminability were not controlled. The difference in RT and performance in attending to auditory items could be related to CI users' poor discriminability of auditory stimuli used in this study. Although we attempted to equate the discriminability of the stimuli between CI and NH listeners with the unimodal task, a fully equal ability to discriminate between two auditory items could not be confirmed. However, once the effects of auditory discriminability on performance and RT were controlled statistically, CI users performed as accurately and as fast as NH listeners. This suggests that although CI users may require a greater perceptual difference between two auditory items to discriminate the two, they can redirect attention to auditory working memory with similar processing speed and accuracy as NH.

Consistent with the present study findings, Luo et al. [43] found similar cuing effects in older CI users and older adults with mild to moderate hearing loss. There was, however, a difference arising from age rather than CI-related declines in working memory and attention when comparing cuing effects with young normal hearing listeners [43]. The lack of group difference in reflective attention observed in the current study may be explained by the prediction that attention to memory deficits are more age-related than hearing-related. Luo et al. [43] also demonstrated greater recall for auditory than visual items, regardless of the cueing condition. Similarly, the present study also showed greater performance and RT for auditory items than visual, regardless of the cueing condition, but only for NH adults. Auditory and visual items were not perceptually equated in the present study, and therefore it is possible that the auditory stimuli were more salient than the visual items, resulting in greater accuracy and RT.

## Visual performance in CI users

Surprisingly, unlike NH controls, CI users demonstrated no significant difference between auditory and visual conditions in variability of response time or accuracy; however, the results trended towards faster and more accurate recall of visual items. Interestingly, CI users show faster responses for visual than auditory items. These findings could be a result of neuroplastic changes related to hearing loss. Given its lack of use, the cortical representation of a deprived sensory system undergoes shrinkage, along with a takeover by the remaining sensory modalities [44]. Research has shown that an absence of auditory input over time can lead to cross-modal plasticity in reorganizing the auditory brain areas for visual processing [45, 46]. The brain tends to perform at supranormal levels in one or more of the intact sensory systems, particularly in the visual system, resulting in compensatory visual function [45]. Prior research on congenitally deaf individuals has shown that the auditory cortex is more activated for visual processing than NH listeners [47–53]. Although limited, current work on post lingually deaf

individuals has also reported evidence for visual take-over of the auditory cortex [54]. More-over, this cross-modal plasticity is experienced not only by deaf individuals but also by blind individuals who compensate for their sensory loss through the remaining intact senses [55]. The brain can rewire its components due to challenges such as sensory deprivation, aiming to adapt and achieve optimal performance using the information from the intact sensory system.

CI users undergo prolonged periods of severe to profound deafness and can exhibit compensatory visual behaviors, but the ability of CIs to reverse cross-modal plasticity is still being explored. CIs produce coarse sound representations, and thus partial hearing recovery may result in an incomplete reversal of cortical changes induced by hearing loss. Although studies have indicated greater responses in the auditory cortex during auditory processing post-implantation, cortical responses differ among CI users [56, 57]. Regarding cross-modal reorganization, studies have found greater activation in the auditory cortex of CI users than NH when processing visual stimuli [58–62]. Cross-modal reorganization of the auditory cortex for visual processing may not be reversed after CI use. This activation pattern has been found to be positively related to lip-reading skills, providing possible functionality of the incomplete reversal of cross-modal reorganization of the auditory cortex [61]. This continued cortical reorganization is beneficial for communication in CI users as it enhances the visual component of speech [63, 64].

Interestingly, some studies have revealed possible visual cortex activation during auditory processing in CI users, even greater visual activation compared to NH listeners [57, 58]. Song et al. [21] found greater visual activation during the audiovisual speech, and weaker activation of auditory brain areas, such as the posterior superior temporal sulcus, during auditory only conditions in CI users. These findings suggest cross-modal visual cortex activation, shedding light on the new patterns of cortical reorganization emerging in CI users. CIs may be unable to reverse cross-modal plasticity once hearing is restored, and perhaps auditory processing exhibits continued reliance on the visual system [21]. This cortical organization has also been positively related to lip-reading abilities, suggesting possible benefits in facilitating communication in CI users [61].

The current study findings indicate that hearing restoration enables CI users to attend to visual items more efficiently, potentially due to continued cross-modal plasticity. This reliance on visual cues may extend to tasks involving retrospective attention, where CI users may benefit from visual representations in memory. Although not significant in this study, the trend suggests a potential role of visual contextual cues in facilitating attentional processes among CI users. The findings of this study suggest that CI users may rely more on visual processing due to continued cross-modal plasticity, highlighting the need for further investigation into the mechanisms underlying sensory reorganization in individuals with hearing loss.

## Limitations

The results suggest that CI users can effectively allocate attention to a representation in auditory and visual working memory. This is unexpected in light of prior research showing that older adults with hearing loss struggle to allocate attention to sound object representation [24]. However, the experimental paradigm used in our study with CI users differed markedly from other paradigms used to investigate reflective attention. One significant difference relates to the number of items participants must hold in working memory. In our CI study, participants held one item in visual and auditory working memory. Prior research used at least two items per modality [24, 25, 41, 42, 65]. The selection of our specific paradigm was influenced by practical considerations inherent to the cochlear implant (CI) user population. Unlike individuals with normal hearing, CI users may face limitations in simultaneously holding multiple

auditory information in working memory. Our simplified paradigm, focusing on a single auditory and visual item in working memory, allowed us to effectively assess the basic level of attentional allocation in CI users. Therefore, the discrepancy between our findings in CI users and those from our previous aging study [24] could be related to the experimental design. It is still being determined whether CI users would also show comparable benefits from the retro-cue if the task involved more than one item in auditory working memory. Increasing the number of items in working memory likely engages a "search" through memory and a selection process, which were unnecessary to perform our simplified version of the delayed match-to-sample task.

In addition, it is important to acknowledge the potential limitations inherent to the design and methodology of this study. Firstly, the sample size of our study may have been insufficient to detect subtle differences or effects, particularly given the heterogeneity within CI users. A larger sample size would enhance the statistical power of our analysis and provide more robust conclusions. Additionally, the absence of equivalence testing in our study design is noteworthy. Equivalence testing would have allowed for a more comprehensive examination of the similarities or differences between CI users and NH peers. Future research should aim to address these methodological limitations to ensure more conclusive and generalizable results.

## Conclusions

The ability to reflect attention to a visual or auditory item in memory, or in other words, refresh memory of a just heard or just seen item, is not affected by the low spectral-temporal resolution of CIs. CI users can refresh their memory in a way similar to that of typical hearing age-matched adults. This ability is at the normal hearing level after one year of implantation and does not differ with greater experience of CI. This was evident as CI users could benefit from a retro-cue that prioritizes a just-remembered item by attending to the cued sensory modality, either visual or auditory. More importantly, the degraded sound information processed by CI users does not limit their ability to attend to an auditory item in their working memory, provided that the auditory stimulus is clearly distinguishable from the item in memory. Future studies should expand this finding by exploring the ability to search and select items in memory using larger set sizes of items. Electrophysiological methods should be employed in these studies to determine if neural processing during reflective attention is affected by hearing acuity when behavioral measures demonstrate comparable performance between groups, as seen in previous findings [66]. Studies should continue to investigate the association of hearing and cognition to help identify processes that would benefit from auditory rehabilitation with the overall aim of improving communication in those who are hard of hearing.

## Supporting information

**S1 Table. Demographic of CI participants including implanted side, CI duration, MoCA and QSIN scores.**
(TIFF)

**S2 Table. Demographic of NH participants including MoCA, QSIN and PTA scores.**
(TIFF)

**S1 Data. Study data.**
(ZIP)

## Acknowledgments

We would like to express our gratitude to Dr. Karen Gordon and Dr. Jay Pratt for their invaluable suggestions on experimental design and data interpretation, which greatly enriched the quality of this manuscript. We also thank Madison Grassi and Mary O'Neil for their diligent efforts in participant recruitment. Additionally, we acknowledge Madison Grassi for her assistance in data collection.

## Author Contributions

**Conceptualization:** Andrew Dimitrijevic, Claude Alain.

**Data curation:** Amisha Ojha.

**Formal analysis:** Amisha Ojha, Andrew Dimitrijevic, Claude Alain.

**Funding acquisition:** Amisha Ojha, Claude Alain.

**Investigation:** Amisha Ojha, Andrew Dimitrijevic, Claude Alain.

**Methodology:** Amisha Ojha, Andrew Dimitrijevic, Claude Alain.

**Project administration:** Amisha Ojha.

**Resources:** Andrew Dimitrijevic, Claude Alain.

**Software:** Amisha Ojha.

**Supervision:** Andrew Dimitrijevic, Claude Alain.

**Validation:** Andrew Dimitrijevic, Claude Alain.

**Visualization:** Amisha Ojha.

**Writing – original draft:** Amisha Ojha.

**Writing – review & editing:** Amisha Ojha, Andrew Dimitrijevic, Claude Alain.

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
