## [Decision Letter · Decision Letter 0]

13 Mar 2024

PONE-D-23-41953Orienting Attention to Auditory and Visual Working Memory in Older Adults with Cochlear ImplantsPLOS ONE

Dear Dr. Ojha,

Thank you for submitting your manuscript to PLOS ONE. After careful consideration, we feel that it has merit but does not fully meet PLOS ONE’s publication criteria as it currently stands. Therefore, we invite you to submit a revised version of the manuscript that addresses the points raised during the review process.

We look forward to receiving your revised manuscript.

Kind regards,

Jyrki Ahveninen

Academic Editor

PLOS ONE

Journal Requirements:

This research was supported by grants from the Natural Sciences and Engineering Research Council of Canada (NSERC) Discovery and the William Demant Foundation to Claude Alain [RGPIN- 2021-02721] and NSERC-Create Training Program in Complex Dynamics grant to Amisha Ojha.

This research was supported by grants from the Natural Sciences and Engineering Research Council of Canada (NSERC) Discovery and the William Demant Foundation to Claude Alain [RGPIN- 2021-02721] and NSERC-Create Training Program in Complex Dynamics grant to Amisha Ojha.

This research was supported by grants from the Natural Sciences and Engineering Research Council of Canada (NSERC) Discovery and the William Demant Foundation to Claude Alain [RGPIN- 2021-02721] and NSERC-Create Training Program in Complex Dynamics grant to Amisha Ojha.

Reviewers' comments:

Reviewer's Responses to Questions

**Comments to the Author**

1. Is the manuscript technically sound, and do the data support the conclusions?

Reviewer #1: Yes

Reviewer #2: No

Reviewer #3: No

2. Has the statistical analysis been performed appropriately and rigorously? 

Reviewer #1: Yes

Reviewer #2: No

Reviewer #3: Yes

3. Have the authors made all data underlying the findings in their manuscript fully available?

Reviewer #1: Yes

Reviewer #2: No

Reviewer #3: No

4. Is the manuscript presented in an intelligible fashion and written in standard English?

Reviewer #1: Yes

Reviewer #2: Yes

Reviewer #3: Yes

5. Review Comments to the Author

Reviewer #1: The study investigates the cue orientation and reflective attention performance of cochlear implant (CI) users within a working memory (WM) task compared to healthy controls. They show that Although in general auditory working memory performance the control group are significantly better than the CI group, there is no significant difference between groups for the retro-cue performance for either auditory or visual WM. Their results suggest that CI users are able to orient their memory similar to typical hearing age matched adults.

The introduction was clear with well explained. The Methods partn needs a bit more details but in general well explained. Results explained in detail clearly. Discussion was read well. I have some comments below:

Methods

Experimental procedure (pp 136-141): Prior to beginning the experiment, participants completed different versions of unimodal (auditory and visual) delayed match-to-sample tasks to familiarize the participants with the paradigm and to adjust the level of discriminability

What is the number of participants in each group taking which version of the unimodal tasks? Were the number of participants taking each unimodal task balanced for each group?

pp 148-151: The physical difference between the gradients (i.e., visual angle rotation) and piano tone (i.e., frequency) varied between versions of the tasks. The difference in visual angle between gradients ranged from a random rotation of 25 to 60 degrees for the visual delayed match-to sample task. For the auditory delayed match-to-sample, the difference in frequency varied from one-quarter tone to seven semitones.

Are these differences counterbalanced within participant? How many trials were there for this task? Were the total number of trials for these tasks same for each participant? Otherwise, there might be a learning effect confound. Are these tasks used a version of a JND to equate the difficulty of the main task for each participant? Is so, are both tasks performed by all participants? This part is unclear in the methods section.

pp. 165-170: Once the stimuli disappeared, a black screen was shown for 1000ms, followed by another 500ms presentation of a letter retro-cue which was either neutral (X) or informative (A or V to indicate whether to orient to auditory or visual modality, respectively). In trials with the informative A retro-cue, participants were tasked to remember the piano tone, whereas trials with the informative V retro-cue, participants were instructed to remember the visual gradients.

What is the instruction for neutral (X) trials?

p 170: …. a test item was presented for 500ms, and participants indicated whether the item matched with the auditory

Is the test item unimodal? What is the probe for each condition? In neutral condition, if the probe is bimodal, what are ‘different’ probe conditions (always both sensory modalities different?)

Results

Does the duration of CI usage have an effect on the WM performance?

Discussion

The unimodal DMTS task suggests a possibility of discrimination problem instead of a WM performance difference in the CI users. This should be discussed a bit more in the paper.

Reviewer #2: While the exploration of the impact of cochlear implants (CIs) on cognitive abilities, particularly on attention orientation towards auditory and visual stimuli in working memory, is commendable and addresses a gap in the literature, several major concerns compromise the integrity and contribution of this study. These issues span the clarity of writing, experimental design, data presentation, and the validity of the conclusions drawn.

Major Concerns:

1. Clarity and Consistency: The manuscript suffers significantly from clarity issues. Terms like “a cue oriented attention” suggest certainty about internal cognitive processes that the study's design cannot conclusively demonstrate. The use of varying terms to describe similar concepts without clear definitions adds to the confusion. For instance, the introduction and abstract confuse more than they clarify the study's objectives and hypotheses. This lack of clarity extends to the explanation of the "delayed match-to-sample task," which is inadequately explained and inconsistently referred to throughout the paper.

2. Experimental Design and Analysis: The paper raises questions about the robustness of its experimental design and statistical analysis. The concern that CI users' comparability to normal-hearing (NH) individuals might stem from a poorly-powered design rather than a genuine lack of difference is alarming. The absence of equivalence testing to support the non-significant results weakens the paper's conclusions. Moreover, the manuscript does not adequately address how variability in CI outcomes—potentially influenced by cognitive abilities—might skew its findings. Such variability is a critical factor that must be controlled or accounted for in the study design.

3. Data Availability and Presentation: The lack of raw data availability hinders the transparency and reproducibility of the findings. Relying solely on highly processed data presented in figures restricts the ability for independent verification and detailed analysis by readers.

4. Interpretation of Findings and Conclusion: The manuscript's final conclusions overreach the data presented. Specifically, claiming stability in reflective attention for CI users without longitudinal data pre- and post-implantation is misleading. Additionally, the assertion of equivalency in performance between CI users and NH individuals is not convincingly supported by the data, especially given the study's design and statistical power concerns.

Minor Concerns:

- The introduction lacks precise references for key claims, such as the variability in outcome measures and the relationship between cognition and CI outcomes. Theoretical frameworks like the sensory deprivation theory are mentioned without sufficient elaboration on their relevance to the study's hypotheses.

- Terms like "excessive slow responses" and abbreviations like "MCI" are introduced without definitions, which could confuse readers unfamiliar with the specific research context.

Recommendations:

1. Improve Clarity: Revise the manuscript to enhance the clarity of the writing. Clearly define all terms and ensure consistency in their use throughout the paper. Provide a detailed explanation of the experimental tasks and hypotheses.

2. Strengthen Experimental Design: Consider additional analyses or follow-up studies that can address the concerns regarding statistical power and equivalence testing. Clarify how the study controls for or accounts for the variability in CI outcomes.

3. Enhance Data Transparency: Make raw data available in a suitable repository to allow for independent verification and analysis. This step is crucial for advancing the field and ensuring the reliability of the findings.

4. Revise Conclusions: Carefully align the conclusions with the data and analyses presented. Avoid overreaching statements that are not directly supported by the study's design or findings.

This study tackles a vital question in the intersection of audiology and cognitive science. However, addressing these concerns is essential for accurately contributing to the understanding of how cochlear implants impact cognitive functions, particularly in the domain of attention and working memory.

Specific comments

Abstract and General Comments:

- The phrase "a cue oriented attention" suggests a certainty about internal cognitive processes not directly observable in the study's design. Suggest rephrasing to accurately reflect the experimental instructions given to participants.

- Clarify the explanation of "delayed match-to-sample task" and ensure consistent terminology throughout the paper.

- Concerns about the study's power to detect differences or equivalences between groups. Consider discussing the possibility of a poorly powered design and the absence of equivalence testing.

- The conclusion regarding "reflective attention to an auditory or visual item in memory remains stable in CI users" is unsupported due to the lack of longitudinal data. Suggest removing or significantly revising this claim.

Introduction:

- Provide references for "variability in outcome measures" and the relationship between cognition and CI outcomes.

- The discussion on how hearing restoration affects cognition lacks clarity and consistency. Clarify these statements and provide evidence or reconsider the framing.

- The study does not investigate brain reorganization directly nor longitudinally; thus, claims about brain reorganization need to be reframed or removed.

- Consider replacing "diminished hearing acuity" with a more precise term like "hearing impairment."

-Change “hypothesized” to “predicted”

- Expand on the sensory deprivation theory and its relevance to your hypotheses. Clarify the study's predictions and consider alternative hypotheses regarding the impact of CI use on cognition. For example: cognition is influenced by hearing impairment, but as implantation restores hearing this also restores cognition. Prediction if this hypothesis is true: no difference.

Methods:

- Define abbreviations like "M" (mean) and "SD" (standard deviation) upon first use.

- Address how age-related hearing loss (presbycusis) in the NH group might affect performance and cognition in the context of your study.

- The "a priori power analysis" section needs clarification, including the rationale behind the expected effect sizes and sample size calculation. Do you consider two groups, two modalities, accuracy and response times? Do you account for the fact that you removed some participants from further analysis (the p value is affected by intention)? What kind of effect size do you expect? Why would this be smaller? Is N = 52 for 2 equal-sized groups? Does N=61 really improve the power enough given a smaller effect size?

- Question the perception of binaural stimuli in CI users and suggest rephrasing or removing misleading terms.

- Clarify the statement that "all stimuli are presented on a computer screen" to accurately reflect how auditory stimuli are presented.

- Discuss the method used to remove outliers (non-recursive moving criterion method), including its impact on data analysis and mean estimation.

Results:

- Consider performing an equivalence test or obtaining a Bayes Factor for non-significant results to more robustly determine the similarity between CI users and NH listeners.

Reviewer #3: Major Comments

1. To evaluate the hypothesis one would prefer the pre and post implantation data. Why did not the authors consider that ?

2. I am not sure the use of wording “reverse” is correct in this context, one can assume that the hearing restoration might compensate for some of those losses but we do not know if it actually reverses or other mechanisms are involved.

3. I do not understand why the slow responders were removed from the analysis?

4. Was there difference between the bilateral and unilateral CI users ?

5. What was the duration of deafness of these CI participants. Does that correlate to their performance?

6. Why was the age not a significant predictor of speech perception and visual retro cue for CI users ?how do authors wrote this?

7. Authors write “Hearing restoration of severe to profound hearing loss by at least one year of CI use yields an ability to attend to items in memory at the normal hearing level” May be I missed this but which result shows this specifically?

8. Given the slight difference in paradigm or sample changes the results completely (e.g. referring to the comparison to findings from Garami et al. and others) how can authors justify that their effect is most relevant to the research question? Or for example what makes their study design better than Garami et al. ?

9. Writing it in the limitations that the paradigm difference made all the difference in this study compared to other might not be convincing enough without a good justification of why the specific paradigm was chosen at the first place?

Minor Comments

1. Table legend needed

2. There are several grammatical errors throughout the manuscript, please consider editing.

6. PLOS authors have the option to publish the peer review history of their article (what does this mean?). If published, this will include your full peer review and any attached files.

Reviewer #1: **Yes: **Isil Uluc

Reviewer #2: No

Reviewer #3: No

---

## [Author Response · Author response to Decision Letter 0]

16 May 2024

We thank the three reviewers for their constructive comments and insightful feedback. Below is our point-by-point response to each comment.

Reviewer #1:

The study investigates the cue orientation and reflective attention performance of cochlear implant (CI) users within a working memory (WM) task compared to healthy controls. They show that although in general auditory working memory performance the control group are significantly better than the CI group, there is no significant difference between groups for the retro-cue performance for either auditory or visual WM. Their results suggest that CI users are able to orient their memory similar to typical hearing age matched adults.

The introduction was clear with well explained. The Methods part needs a bit more details but in general well explained. Results explained in detail clearly. Discussion was read well. I have some comments below:

Methods Experimental procedure (pp 136-141): Prior to beginning the experiment, participants completed different versions of unimodal (auditory and visual) delayed match-to-sample tasks to familiarize the participants with the paradigm and to adjust the level of discriminability

What is the number of participants in each group taking which version of the unimodal tasks? Were the number of participants taking each unimodal task balanced for each group?

RESPONSE: We thank the reviewer for pointing that out. We revised the method section and now clarify that all participants in both groups completed both the auditory and visual unimodal tasks, which began with an initial level of two semitones and a 30-degree angle difference between stimuli. We also indicate in the results section the number of participants taking which version of each unimodal task.

pp 148-151: The physical difference between the gradients (i.e., visual angle rotation) and piano tone (i.e., frequency) varied between versions of the tasks. The difference in visual angle between gradients ranged from a random rotation of 25 to 60 degrees for the visual delayed match-to sample task. For the auditory delayed match-to-sample, the difference in frequency varied from one-quarter tone to seven semitones.

Are these differences counterbalanced within participant? How many trials were there for this task? Were the total number of trials for these tasks same for each participant? Otherwise, there might be a learning effect confound. Are these tasks used a version of a JND to equate the difficulty of the main task for each participant? Is so, are both tasks performed by all participants? This part is unclear in the methods section.

RESPONSE: All participants began the familiarization phase with a difference of a 30-degree visual angle rotation and two semitones for the visual and auditory delayed match-to-sample, respectively. They completed a block of 14 trials. If performance was lower than 85%, the physical difference was increased. The difference was reduced if performance was at the ceiling (i.e., 100% accuracy). This process was repeated until performance was approximately 85% accurate in auditory and visual working memory tasks. For each participant, the auditory and visual stimuli that yielded approximately 85% accuracy in the unimodal task were then incorporated into the audio-visual delayed match-to-sample task. 

pp. 165-170: Once the stimuli disappeared, a black screen was shown for 1000ms, followed by another 500ms presentation of a letter retro-cue which was either neutral (X) or informative (A or V to indicate whether to orient to auditory or visual modality, respectively). In trials with the informative A retro-cue, participants were tasked to remember the piano tone, whereas trials with the informative V retro-cue, participants were instructed to remember the visual gradients.

What is the instruction for neutral (X) trials?

RESPONSE: The participants were told that the letter X was uninformative and that the probe could be visual or auditory. This has been clarified in the method section.

p 170: …. a test item was presented for 500ms, and participants indicated whether the item matched with the auditory

Is the test item unimodal? What is the probe for each condition? In neutral condition, if the probe is bimodal, what are ‘different’ probe conditions (always both sensory modalities different?)

RESPONSE: We thank the reviewer for this comment. Yes, the test item is unimodal. The probe item (S2) is similar to S1, only differing in either pitch or visual angle. This has been clarified in the method section.

Results Does the duration of CI usage have an effect on the WM performance?

RESPONSE: In our sample, the correlation between CI duration and retro-cue benefit in RT for either modality was not significant. This is now clarified in the results section.

Discussion The unimodal DMTS task suggests a possibility of discrimination problem instead of a WM performance difference in the CI users. This should be discussed a bit more in the paper.

RESPONSE: We thank the reviewer for this comment and have added a paragraph to address the discrimination problem in the discussion section. We highlight that although CI users may require a greater perceptual difference between two auditory items to discriminate them, they can orient attention to auditory working memory with similar processing speed and accuracy as NH.

Reviewer #2:

While the exploration of the impact of cochlear implants (CIs) on cognitive abilities, particularly on attention orientation towards auditory and visual stimuli in working memory, is commendable and addresses a gap in the literature, several major concerns compromise the integrity and contribution of this study. These issues span the clarity of writing, experimental design, data presentation, and the validity of the conclusions drawn.

Major Concerns:

Clarity and Consistency: The manuscript suffers significantly from clarity issues. Terms like “a cue oriented attention” suggest certainty about internal cognitive processes that the study's design cannot conclusively demonstrate. The use of varying terms to describe similar concepts

without clear definitions adds to the confusion. For instance, the introduction and abstract confuse more than they clarify the study's objectives and hypotheses. This lack of clarity extends to the explanation of the "delayed match-to-sample task," which is inadequately explained and inconsistently referred to throughout the paper.

RESPONSE: We appreciate the reviewer’s concerns and have made substantial revisions to all sections. We have clarified several terms and, on occasion, provided examples. We hope that the revised document is clearer. We have revised the description of the delayed match-to-sample task.

Experimental Design and Analysis: The paper raises questions about the robustness of its experimental design and statistical analysis. The concern that CI users' comparability to normal-hearing (NH) individuals might stem from a poorly-powered design rather than a genuine lack of difference is alarming. The absence of equivalence testing to support the non-significant results weakens the paper's conclusions. Moreover, the manuscript does not adequately address how variability in CI outcomes—potentially influenced by cognitive abilities—might skew its findings. Such variability is a critical factor that must be controlled or accounted for in the study design.

RESPONSE: We acknowledge the study's limitations. Nonetheless, the present study provides important findings to guide further research. We have revised the section on the sample size and provided evidence supporting our claim that we have sufficient power to detect group differences. We agree that performance was not “equivalent” and aimed to statistically account for the “absence of equivalence.” We have expanded on these caveats in the discussion. We agree with the reviewer that there are likely other cognitive factors not captured in our testing. 

This is now mentioned in the intro and also mentioned in the discussion (visual performance in CI). 

Data Availability and Presentation: The lack of raw data availability hinders the transparency and reproducibility of the findings. Relying solely on highly processed data presented in figures restricts the ability for independent verification and detailed analysis by readers.

RESPONSE: We agree and will make some data available according to the guidelines from PLoS. Regrettably, not all data could be included in this submission as it contains Personal Health Information, as mandated by the Research Ethics Board at Baycrest Health Sciences. However, researchers interested in accessing the data may do so by contacting the ethics committee through Koblinsky at 416-785-2500 ext. 2440 or Ashley Kim at 416-785-2500 ext. 3550.

Interpretation of Findings and Conclusion: The manuscript's final conclusions overreach the data presented. Specifically, claiming stability in reflective attention for CI users without longitudinal data pre- and post-implantation is misleading. Additionally, the assertion of equivalency in performance between CI users and NH individuals is not convincingly supported by the data, especially given the study's design and statistical power concerns.

RESPONSE: We agree with the reviewer that the claim of “stability” should ideally be tested in pre-post longitudinal experimental designs. We have clarified this in the discussion. The sample size and characterization of our sample provide sufficient statistical power. The cross-sectional design is warranted as a first step to examine differences between populations. We have revised the discussion section, limitation, and conclusion sections. We feel that our study provides a solid foundation upon which further studies of reflective attention in CI users could be developed.

Minor Concerns:

The introduction lacks precise references for key claims, such as the variability in outcome measures and the relationship between cognition and CI outcomes. Theoretical frameworks like the sensory deprivation theory are mentioned without sufficient elaboration on their relevance to the study's hypotheses.

RESPONSE: We have revised the introduction and discussed the sensory deprivation theory in more detail and how it relates to our study.

Terms like "excessive slow responses" and abbreviations like "MCI" are introduced without definitions, which could confuse readers unfamiliar with the specific research context.

RESPONSE: Thank you for bringing this to our attention. We have defined all abbreviations and terms. 

Recommendations:

Improve Clarity: Revise the manuscript to enhance the clarity of the writing. Clearly define all terms and ensure consistency in their use throughout the paper. Provide a detailed explanation of the experimental tasks and hypotheses.

RESPONSE: We have revised the manuscript and enhanced the clarity of the writing. We have clearly defined all terms and ensured consistency throughout the paper. The experimental tasks and hypotheses are explained in detail.

Strengthen Experimental Design: Consider additional analyses or follow-up studies that can address the concerns regarding statistical power and equivalence testing. Clarify how the study controls for or accounts for the variability in CI outcomes.

RESPONSE: Thank you for this suggestion. We have expanded to include topics related to “equivalence testing” and confounding factors. 

Enhance Data Transparency: Make raw data available in a suitable repository to allow for independent verification and analysis. This step is crucial for advancing the field and ensuring the reliability of the findings.

RESPONSE: The raw data will be made available once the manuscript is accepted for publication.

Revise Conclusions: Carefully align the conclusions with the data and analyses presented. Avoid overreaching statements that are not directly supported by the study's design or findings.

RESPONSE: The conclusion has been significantly revised.

This study tackles a vital question in the intersection of audiology and cognitive science. However, addressing these concerns is essential for accurately contributing to the understanding of how cochlear implants impact cognitive functions, particularly in the domain of attention and working memory.

RESPONSE: We agree and have made several changes to address the reviewer’s comments and suggestions.

Specific comments:

The phrase "a cue oriented attention" suggests a certainty about internal cognitive processes not directly observable in the study's design. Suggest rephrasing to accurately reflect the experimental instructions given to participants.

RESPONSE: We thank the reviewer for bringing this to our attention. “cue oriented attention” is a commonly used term in the psychology literature. We have added more clarification.

Clarify the explanation of "delayed match-to-sample task" and ensure consistent terminology throughout the paper.

RESPONSE: We have clarified the experimental design and improved consistency of terminology throughout the manuscript.

Concerns about the study's power to detect differences or equivalences between groups. Consider discussing the possibility of a poorly powered design and the absence of equivalence testing.

RESPONSE: We have expanded on the caveats of the study in our discussion.

The conclusion regarding "reflective attention to an auditory or visual item in memory remains stable in CI users" is unsupported due to the lack of longitudinal data. Suggest removing or significantly revising this claim.

RESPONSE: The conclusion section has been revised.

Introduction: Provide references for "variability in outcome measures" and the relationship between cognition and CI outcomes.

RESPONSE: The introduction has been revised and now includes more references for variability outcome measures of CI users. The relationship between CI use and cognition has also been included. 

The discussion on how hearing restoration affects cognition lacks clarity and consistency. Clarify these statements and provide evidence or reconsider the framing.

RESPONSE: The introduction includes a section discussing the effects of hearing restoration with CI on cognition. We highlight that the findings on the benefits of CI on cognition are mixed.

The study does not investigate brain reorganization directly nor longitudinally; thus, claims about brain reorganization need to be reframed or removed.

RESPONSE: We thank the reviewer for this comment. In the present study, we mention cross-modal plasticity as a possible explanation for performance in CI users. We do not claim that our findings support or do not support cross-modal plasticity but rather acknowledge that this could account for some of our findings. This section has been revised to reflect that.

Consider replacing "diminished hearing acuity" with a more precise term like "hearing impairment."

RESPONSE: This has now been revised.

Change “hypothesized” to “predicted”

RESPONSE: This has now been revised.

Expand on the sensory deprivation theory and its relevance to your hypotheses. Clarify the study's predictions and consider alternative hypotheses regarding the impact of CI use on cognition. For example: cognition is influenced by hearing impairment, but as implantation restores hearing this also restores cognition. Prediction if this hypothesis is true: no difference.

RESPONSE: The sensory deprivation theory and how it relates to our hypothesis has been expanded in the introduction.

Methods: Define abbreviations like "M" (mean) and "SD" (standard deviation) upon first use.

RESPONSE: We thank the reviewer for pointing that out. We have revised the text and now define the abbreviations. 

Address how age-related hearing loss (presbycusis) in the NH group might affect performance and cognition in the context of your study.

RESPONSE: A description of presbycusis and how it may affect cognition has been added to the introduction.

The "a priori power analysis" section needs clarification, including the rationale behind the expected effect sizes and sample size calculation. Do you consider two groups, two modalities, accuracy and response times? Do you account for the fact that you removed some participants from further analysis (the p value is affected by intention)? What kind of effect size do you expect? W

---

## [Decision Letter · Decision Letter 1]

1 Aug 2024

PONE-D-23-41953R1Orienting Attention to Auditory and Visual Working Memory in Older Adults with Cochlear ImplantsPLOS ONE

Dear Dr. Ojha,

Thank you for submitting your manuscript to PLOS ONE. We feel that it has merit but does not yet fully meet PLOS ONE’s publication criteria as it currently stands. Therefore, we invite you to submit a revised version of the manuscript that addresses the remaining points raised by Reviewer 1.

We look forward to receiving your revised manuscript.

Kind regards,

Jyrki Ahveninen

Academic Editor

PLOS ONE

Journal Requirements:

Reviewers' comments:

Reviewer's Responses to Questions

**Comments to the Author**

1. If the authors have adequately addressed your comments raised in a previous round of review and you feel that this manuscript is now acceptable for publication, you may indicate that here to bypass the “Comments to the Author” section, enter your conflict of interest statement in the “Confidential to Editor” section, and submit your "Accept" recommendation.

Reviewer #1: (No Response)

Reviewer #3: All comments have been addressed

2. Is the manuscript technically sound, and do the data support the conclusions?

Reviewer #1: Yes

Reviewer #3: Partly

3. Has the statistical analysis been performed appropriately and rigorously? 

Reviewer #1: Yes

Reviewer #3: Yes

4. Have the authors made all data underlying the findings in their manuscript fully available?

Reviewer #1: Yes

Reviewer #3: No

5. Is the manuscript presented in an intelligible fashion and written in standard English?

Reviewer #1: Yes

Reviewer #3: Yes

6. Review Comments to the Author

Reviewer #1: The task for the uninformative (X) condition is still unclear in the methods section. From what I gathered, they needed to remember both A and V items and respond to a unilateral item but it is still not clear. As this is 50% of the trials. It would be a good idea to clarify. If this is not the case, the Adiovisual delayed match to sample explanation still needs more clarification.

Reviewer #3: Thank you for addressing the comments. I hope subsequent studies will follow these findings to further reinforce these outcomes but for this one I don't have any further comments.

7. PLOS authors have the option to publish the peer review history of their article (what does this mean?). If published, this will include your full peer review and any attached files.

Reviewer #1: **Yes: **Isil Uluc

Reviewer #3: **Yes: **Nabin Koirala

---

## [Author Response · Author response to Decision Letter 1]

21 Aug 2024

We thank the two reviewers for their constructive comments and insightful feedback. Below is our point-by-point response to each comment.

Reviewer #1:

The task for the uninformative (X) condition is still unclear in the methods section. From what I gathered, they needed to remember both A and V items and respond to a unilateral item but it is still not clear. As this is 50% of the trials. It would be a good idea to clarify. If this is not the case, the Audiovisual delayed match to sample explanation still needs more clarification.

RESPONSE: We thank the reviewer for highlighting this point. The reviewer's interpretation is correct. We have revised the methods section to clarify the description of the uninformative (X) condition, which now reads as follows:

“In the neutral X retro-cue condition, participants were required to retain both auditory and visual items in memory, as no indication was provided regarding which modality would be tested. The test item could originate from either modality, necessitating participants to match it with the corresponding item in their working memory. This condition accounted for 50% of the trials and required participants to be prepared for either test item.”

---

## [Editor Report · Decision Letter 2]

26 Aug 2024

Orienting Attention to Auditory and Visual Working Memory in Older Adults with Cochlear Implants

PONE-D-23-41953R2

Dear Dr. Ojha,

We’re pleased to inform you that your manuscript has been judged scientifically suitable for publication and will be formally accepted for publication once it meets all outstanding technical requirements.

Kind regards,

Jyrki Ahveninen

Academic Editor

PLOS ONE
---

## [Editor Report · Acceptance letter]

29 Aug 2024

PONE-D-23-41953R2 

PLOS ONE

Dear Dr. Ojha, 

I'm pleased to inform you that your manuscript has been deemed suitable for publication in PLOS ONE. Congratulations! Your manuscript is now being handed over to our production team.

Kind regards, 

on behalf of

Dr. Jyrki Ahveninen 

Academic Editor

PLOS ONE